# Biological Strategies of Invasive Bark Beetles and Borers Species

**DOI:** 10.3390/insects12040367

**Published:** 2021-04-20

**Authors:** Denis A. Demidko, Natalia N. Demidko, Pavel V. Mikhaylov, Svetlana M. Sultson

**Affiliations:** 1Sukachev Institute of Forest, Siberian Branch, Russian Academy of Science, 50, bil. 28, Akademgorodok, 660036 Krasnoyarsk, Russia; 2Scientific Laboratory of Forest Health, Reshetnev Siberian State University of Science and Technology, Krasnoyarskii Rabochii Prospekt. 31, 660037 Krasnoyarsk, Russia; sultson2011@yandex.ru; 3Department of Medical and Biological Basics of Physical Education and Health Technologies, School of Physical Education, Sport and Tourism, Siberian Federal University, Svobodny ave. 79, 660041 Krasnoyarsk, Russia; bnn77@bk.ru

**Keywords:** biological invasions, bark beetles and borers, biological features, invasion patterns, data mining

## Abstract

**Simple Summary:**

Biological invasions are one of the most critical problems today. Invaders have been damaging tree- and shrub-dominated ecosystems. Among these harmful species, a notable role belongs to bark beetles and borers. Extensive phytosanitary measures are needed to prevent their penetration into new regions. However, the lists of quarantine pests should be reasonably brief for more effective prevention of invasion of potentially harmful insects. Our goal is to reveal the set of biological traits of invasive bark beetles and borers that are currently known. We identified four invasion strategies. *Inbred*, the first one is characterized by inbreeding, parthenogenesis, polyvoltinism, xylomycetophagy, flightless males, polyphagy, to less extent by association with pathogenic fungi. For the second, *polyphagous*, typical traits are polyphagy, feeding on wood, high fecundity, distance sex pheromones presence, development for one year or more. The third strategy, *intermediate*, possesses such features as mono- or olygophagy, feeding on inner-bark, short (one year or less) life cycle. *Aggressive*, the last one includes monophagous species using aggregation pheromones, associated pathogens, short life cycle, and consuming inner-bark. The main traits contributing to significant damage are high fecundity, polyvoltinism, symbiotic plant pathogens, long-range or aggregation pheromones.

**Abstract:**

The present study attempts to identify the biological characteristics of invasive (high-impact in the secondary area) bark beetles and borers species, contributing to their success in an invaded area. We selected 42 species based on the CABI website data on invasive species and information on the most studied regional faunas. Four groups of species with different invasion strategies were identified based on the cluster and factor analysis. The first one (inbred strategy) is characterized by flightless males, xylomycetophagy, low fecundity (~50 eggs), inbreeding, polyvoltinism, and polyphagy. Species with an aggressive strategy are poly- or monovoltine, feeds on a limited number of hosts, larval feeding on the inner bark, are often associated with phytopathogens, and produce aggregation pheromones. Representatives of the polyphagous strategy have a wide range of hosts, high fecundity (~150 eggs), larval feeding on wood, and their life cycle is at least a year long. For the intermediate strategy, the typical life cycle is from a year or less, medium fecundity, feed on inner bark tissues, mono- or oligophagy. Comparison with low-impact alien species showed that the most significant traits from the viewpoint of the potential danger of native plant species are high fecundity, polyvoltinism, presence of symbiotic plant pathogens, long-range or aggregation pheromones.

## 1. Introduction

Biological invasions are one of the most crucial problems facing modern ecologists [1]. Research in this field is relevant because invasive species may cause socio-cultural, economic, and environmental damage or harm human health [2]. Hereafter, we use the term invasive only for alien species, demonstrating economic or ecologic impact in the new regions. If we use the term alien, we mean both invasive and non-invasive species. All terrestrial ecosystems, including biomes dominated by trees and shrubs, both natural and anthropogenic, suffer from invasive species to some extent [3]. 

The level of knowledge on the fauna of adventive species associated with trees and shrubs differs significantly from region to region. However, the available information allows one to get an idea of the contribution of different ecological groups of forest insects to the introduction process. Studies of forest insects in the USA, Europe, China, and Canada have shown that bark beetles and wood borers occupy 10–20% of adventive fauna species [4,5,6]. According to data for the USA and Canada, such insects also account for 12–20% of the species causing significant damage [5,7]. 

Even though bark beetles and borers occupy a relatively small proportion among other invaders damaging trees and shrubs, the damage they cause can be pretty significant. For example, the emerald ash borer *Agrilus planipennis* Fairmaire, a native of the Far East’s broad-leaved forests, damages ash-dominated stands outside its native range. In some United States regions, almost all ash trees with stems greater than 2.5 cm in diameter have had their canopies killed by the emerald ash borer. Both forest stands in settlements and natural forests suffer from this insect [8]. In the European part of Russia, this species is also widespread and highly harmful. For instance, in Moscow, this pest killed about a million ash trees [9]. 

The rate of alien species introductions outside their native range has been increasing since the late 19th–early 20th centuries, showing no signs of slowing down. It makes the threat to tree- and shrub-dominated biocenoses from invading insects even more dangerous. Growth in international trade volume, resulting in the unintentional import of living organisms into new regions, contributes to the increasing threat of invasions [10,11]. Even though international trade volume is forecasted to fall, it is unlikely to seriously slow down the invasion process (the decline for different groups of goods and services world trade is expected to fall by between 1.29% and 9.26%) [12]. Furthermore, the most significant group of goods from the perspective of bark beetles and borers invasion—Forest and Wood Products [10]—will suffer the least [12]. 

Even the implementation of phytosanitary measures in the future is unlikely to prevent the invasion of forest insects into new areas or reduce its rate since increase in global trade growth will offset the success of quarantine measures [11]. Identifying species that pose the most significant potential threat, may be a practical approach. However, the approach is challenging to implement due to a lack of knowledge about bark beetles and bores biological traits [13]. There are features that are considered to determine some species’ invasion success compared to others [11]. In the present research, we tried to identify such biological characteristics for insects that tunnel and feed under the bark in living wood and can cause significant damage in an introduced area.

## 2. Materials and Methods

### 2.1. Sources and Data Collection

The Center for Agriculture and Biosciences International (CABI) website was the source of information on invasive bark beetles and wood borers worldwide [14]. Moreover, we used data on their fauna in the regions where it is represented by a significant number of species and is particularly well studied: The European part of Russia (order Coleoptera only) [9], the USA [5], China [6] and New Zealand [15] (Table 1). In at least one of these or additional sources, the species was listed as causing severe ecological or socio-economic damage. Additionally, we used DAISIE data [16] for European and Mediterranean countries. As DAISIE does not contain any invasiveness information, we used additional sources for this purpose [17,18,19,20]. We included in the research those species that develop beneath the bark or inner tissues of lignified sections of trees and shrubs, including bamboos and palms, but excluding agaves, yuccas, and decaying wood. In total, we examined 47 species of insects.

When determining the type of nutrition, we did not consider the tunneling that was not accompanied by feeding on wood. For example, representatives of the tribe Xyleborini (Curculionidae: Scolytinae) are typical ambrosia beetles that tunnel into the wood to breed but feeding exclusively on their fungal symbiont [53]. Likewise, the larvae of many species of longhorn beetles (Cerambycidae) tunnel within wood before pupation, but do not actually eat it. Besides, the type of nutrition can change at different development stages, so each type was considered a Boolean variable, and the same species can have more than one type of nutrition. 

For each species, we analyzed information about the traits that, according to the literature data [10] or relying on our assumptions, contribute to the invasion success: nutrition (type of nutrition, maturate feeding, food specialization), reproduction (fecundity, parthenogenesis and inbreeding, life cycle), aggregation or long-range sex pheromones, association with pathogens, ability to fly (Table 2). We filled the missing data in the main sources using additional data (Table 1; for more details, see Appendix A).

We considered monophagous species that larval trophic behavior could be associated mainly with one plant genus and oligophages—species that widely use plants within the same family [11]. We referred to insects that typically live on species of more than one family of trees or shrubs as polyphages. The known, but rare, instances of feeding on plants outside the preferred taxon were not taken into account in the case of mono- and oligophagy. For example, the four-eyed bark-beetle *P. proximus*, which has always been considered a monophagous species on the genus *Abies*, in some cases, is still able to successfully reproduce on representatives of other genera of conifers [82]. 

When assessing fecundity, we used the arithmetic mean given in the data source. For the species that fecundity data was presented in several sources, we calculated the arithmetic mean for each separately and then averaged the results. If the fecundity study was carried out in several modes [14,65], we used the results for the conditions closest to natural. 

Insect development rates, as well as those of other poikilothermic animals are dependent upon environmental conditions, mainly temperature [105]. As a result, the species life cycle may vary in different parts of the current range [9,106] and even on the same site in different years [82]. Therefore, we considered voltinism for each species (polyvoltine, monovoltine, and multi-year life cycles) as Boolean variables (as for the type of nutrition). 

The release of aggregation pheromones facilitates individuals’ concentration in a limited area [107], weakening the Allee effect in the early invasion stages. Secondly, aggregation pheromones contribute to the mass attack and colonization of trees, so they are able to overcome tree defense mechanisms [11,53]. The presence of long-range sex pheromones also facilitates the Allee effect to be overcome [108]. We did not include other types of attractants, such as kairomones, as they help bark beetles and borers search for food [109], but do not provide the advantages described above. Contact sex pheromones [110] and sex pheromones of low volatility [103] were not considered in the present study because they influence behavior only at close-range. 

We used the same sources and criteria when selecting non-invasive alien species (Table 3) for further comparison with invasive ones. 

Analyzing data sources, we faced the data scarcity problem indicated in [13]. Therefore, in some cases, we used similar data obtained for closely related species (Table 4). If the information on some traits was absent, the missing values were modeled statistically for further calculations (for more information, see the Statistical processing) (Table 4). In the absence of data on three or more traits, the species were excluded from the research.

### 2.2. Statistical Processing

All calculations were performed in the R 4.0.2 software environment for statistical computing and graphics [138]. A preliminary step was filling in the missing data using the random forest method implemented in the mice 3.11.0 package [139]. Then we calculated a matrix of distances among species using Gower’s distance metric and performed clustering. The primary method of cluster analysis was partitioning around medoids (PAM). We carried out a fuzzy clustering to better understand the structure of the groups obtained. Fuzzy clustering results describe the probability of items falling into the identified clusters [140]. Such data allows one to understand the connections between groups better and consider the features of items located at the boundaries between them. 

As a measure of the PAM between-cluster isolation, we used the silhouette width—a value calculated based on the average within-cluster distance and the average distance to the closest object from the other cluster. The more similar an item is to other within-cluster objects, the wider the silhouette. The high mean arithmetic silhouette index value for all objects included in it (*W*) indicates an isolated cluster. We determined the PAM clustering stability by cluster bootstrapping. Bootstrapping results (*boot*) represent the mean values of the Jaccard similarity coefficient for the initial clusters and clusters constructed after the classification of random samplings with replacement [141]. We used the cluster 2.1.0 package [140] to calculate the distances between species, the silhouette width, and cluster analysis. Bootstrapping estimates of stability for clusters were performed using the fpc 2.2–8 package [141].

The correlation between insects’ biological traits and their belonging to a particular cluster was evaluated using Fisher’s exact test for Boolean traits (almost all traits) or the Kruskal–Wallis non-parametric method for quantitative traits (fecundity). We compared each cluster to the others for all the considered features. In all cases, we considered the differences as significant at *p* ≤ 0.05.

Moreover, we processed materials on the biological attributes of invasive insect pests using factor analysis of mixed data (FAMD) implemented in the FactoMineR 2.3 package [142]. FAMD helped us perform a more accurate interpretation of the traits data obtained in frames of cluster analysis, understand the structure of correlation between these traits and differences between invasive and non-invasive bark beetles and borers species.

## 3. Results

### 3.1. Taxonomic Analysis of Invasive Bark Beetles and Wood Borers

Almost all of the 47 invasive species that we classified as invasive are members of the order Coleoptera 59.6% (28 species) belong to the family Curculionidae (including species of Platypodinae and Dryophthorinae subfamilies, considered as distinct families in some taxonomic systems [53,143,144]). Notably, nine of them belong to one tribe Xyleborini of the subfamily Scolytinae. Another eleven species (23.4%) belong to the Cerambycidae family. The other families have significantly fewer species. Three species (6.4%) belong to the Bostrychidae family (including the Lyctidae family as a subfamily [144]). Two species belong to the Buprestidae family. Other orders are represented only by two species Hymenoptera (family Siricidae) and one species Lepidoptera (family Castniidae) (Table 5).

Generally, the regional faunas of alien bark beetles and borers correspond to the following pattern: Curculionidae represents a significant proportion of species; Cerambycidae takes somewhat less proportion (Table 5). Other families are represented by several species and not in all regions. To a certain extent, the Buprestidae and Siricidae families can be considered an exception to the pattern since its species, in small numbers, are indicated for three out of four local faunas (Table 5). Integrated lists of both invasive and alien bark beetles and borers generalized for four regions [5,6,9,15] also follow this pattern (we excluded DAISIE data due to the lack of information about food guilde and impact of the species).

### 3.2. Characteristics of the Studied Invasive Species

All three types of nutrition are widely represented among the species selected for the research. Feeding on inner-bark is more typical (27 species, 57.4%). Less often, larvae consume wood or mycelium of symbiotic fungi (14 and 15 species, 29.8 and 31.9% respectively). The food spectrum is also uniform; there are 15 (31.9%), 11 (23.4%), and 21 (44.72%) monophagous, oligophagous, and polyphagous species, respectively. Maturation feeding is known for most invasive bark beetles and borers (35 species, 74.5%). Eight species do not feed during the imaginal stage (17.0%). There is no nutrition data for four species available in the known sources. 

A tendency to inbreed was indicated for twelve species (25.5%) considered in the present study. The mean fecundity of invasive bark beetles and borers’ species ranged between 15 and 340 eggs per female. However, in general, fecundity is not high: the median value for all species is 83 eggs; the quartile range is from 25 to 110 eggs.

Among the biological features affecting an insect’s aggression towards plants, the association with pathogens is relatively widespread. It was noted for 16 species (34.0%). There are numerous species using aggregation pheromones (12 species, 25.5%) and long-range sex pheromones (6 species, 12.8%).

The species included in the research, to one degree or another, are capable of active flight. However, for eleven (23.4%) species, wings are present in only females. 

The life cycle of invasive bark beetles and borers can vary from several generations per year to 2–3 years. In general, species with a short life cycle are more widespread: 30 (63.8%) polyvoltine species and 31 monovoltine species (66.0%). However, there still are 16 species (34.0%) developing over many years. 

### 3.3. Clustering

At the preliminary stage of the cluster analysis, we found that identifying four groups is optimal for the best interpretation. Such clustering avoids both the artificial splitting into a large number of slightly different clusters and the unjustified grouping of species into large, internally heterogeneous clusters. Further, for simplicity of presentation, we will use the selected groups and their strategies as synonyms. 

The PAM clustering results are presented in Figure 1. The first of the identified clusters (13 species) was named the *polyphagous*-type. It is not distinctly isolated from other clusters but stable. Another large group, including 10 species (*intermediate*-type), has the smallest mean silhouette width and the stability value. The most isolated and stable cluster (*inbred*-type), which includes almost exclusively Xyleborini species, also consists of 13 ones. The *aggressive*-type cluster (11 species) is poorly isolated from the others and is the least stable (excluding the *intermediate* group). The presence in all clusters of species with small, up to negative silhouette width values indicates their intermediate position (see Biological characteristics of the identified strategies).

Before carrying out fuzzy clustering, we considered the issue of choosing the value of the parameter memb.exp (membership exponent), which determines the fuzziness of the membership assignments of the clustering. A deviation from the optimal value could lead, either to an almost equiprobable inclusion of a species in all clusters, or conversely, to an almost 100% probability of its assignment to only one cluster. The results obtained would obviously not make sense in either case. The best interpretation was achieved at memb.exp = 1.8. 

The results of fuzzy clustering indicated significant heterogeneity of most of the identified clusters (Appendix A). The probabilities of cluster membership, calculated based on fuzzy clustering, confirmed the *inbred*-type homogeneity and its distinct isolation from other groups. The probability of attributing its representatives to other clusters exceed 0.2 only for two species. Notably, this applies to species other than the tribe Xyleborini (*D. micans* and *T. fuscicornis*). Both species classified as *polyphagous* and *aggressive* types tend to *intermediate*-type traits. To a lesser extent, invasive bark beetles and borers were characterized by a set of features intervening between the *polyphagous* and *intermediate* strategies. Thus, in some cases, species have transitional sets of traits between the selected types.

### 3.4. Biological Characteristics of the Identified Strategies

Among the identified invasion strategies, the *inbred*-type is distinguished by the most apparent set of features. The following traits characterize the species of this group: males of most species are unable to fly, the absolute absence of aggregative pheromones and long-range sex pheromones (Figure 2), xylomycetophagy (Figure 3), inbreeding, and low fecundity (Figure 4). The *inbred*-type is close to some other strategies regarding polyvoltinism (Figure 4) and trophic diversity (Figure 3). Remarkably, a relatively high proportion of this group species is associated with pathogens (Figure 4).

The following traits characterize the species of the *aggressive*-type strategy: A short life cycle, medium fecundity values (Figure 2), narrow food specialization, phleophagy (Figure 3), most species are pathogen-associated and produce aggregation pheromones (Figure 2).

The following traits characterize the species of the *aggressive*-type strategy: A short life cycle, medium fecundity values (Figure 4), narrow food specialization, phleophagy (Figure 3), most species are pathogen-associated and produce aggregation pheromones (Figure 2). 

The following traits characterize the species representing the *polyphagous*-type strategy: A wide range of host species, a significant proportion of species feeding exclusively or partially on wood at the larval stage (Figure 3), high mean fecundity, and long life cycle (Figure 4). A significant proportion of species do not feed during the imaginal stage (Figure 3) and produce long-range sex pheromones (Figure 2).

It is worth emphasizing that there is a high probability that some species are included in certain groups due to insufficient knowledge. For example, *P. archon* is supposed to produce aggregation pheromones [32]. The aggregation pheromones are probably used by *D. minutus* since it was discovered in the closely related *Dinoderus bifolveolatus* Woll. [145]. Association with plant pathogens (rust fungi) is possible for *L. festiva* [30]. Notably, the possibility of spreading Dutch elm disease was shown for *S. schevyrevi* [87], located between *aggressive* and *intermediate* types according to the available data.

Factor analysis results correspond well to the data obtained by comparing the selected groups. The first three dimensions are well interpreted in the resulting feature space, explaining about 60% of the total variance. The first dimension distinguished those characterized by xylomycetophagy and its attendant features (*inbred*-type) and fecundity (Figure 5; see also Appendix A and Appendix A). The second dimension has a complex nature: the coordinates on the corresponding axis are determined by food specialization (monophagy/polyphagia), type of nutrition, life cycle duration, and the presence of pheromones (Figure 5A; see also Appendix A and Appendix A). The third dimension separates oligophagous species from other ones. Monovoltine species and ones with symbiotic plant pathogens or long-range sex pheromones are located on the opposite end of this axis (Figure 5B; see also Appendix A and Appendix A).

### 3.5. Invasive vs. Non-Invasive Species Biological Traits

When we performed the factor analysis for invasive and non-invasive species together using invasiveness as a trait, it showed a high contribution of invasiveness to the third, fourth, and sixth dimensions (which explained 23.4% of total variance). Apart from invasiveness, the following traits turned out to be closely related with these dimensions: Presence of long-range sex pheromones and absence of maturation feeding (third dimension), presence of aggregation pheromones (fourth dimension), polyvoltinism (sixth dimension), presence of symbiotic plant pathogens (third and fourth), fecundity (third and sixth), and host plant specificity (fourth and sixth) (Figure 6; see also Appendix A and Appendix A).

## 4. Discussion

### 4.1. Taxonomic Structure of Invasive Species

The analysis of the representation of various bark beetles and borers families among invaders revealed that their taxonomic features are generally close to those for the entire set of alien species of the same food guild (Table 5). Feeding on the tissues of stems or branches of trees and shrubs is typical for most species (or at least a significant number of them) of the families listed as invaders (Table 1). However, the size of these families is very uneven. Only 570 species belong to the Bostrichidae family, while the rest are among the more prosperous taxa. A total of 14,700 species belong to the Buprestidae, 30,000 species to the Cerambycidae, and 51,000 species to the Curculionidae [143]. Remarkably, most of the Curculionidae family bark beetles and borers belong to the subfamily Scolytinae (5990 species) [53], which includes all but three species among invaders from the Curculionidae family. Families from other orders, Siricidae and Castniidae, are small: the first has 122 species [137], the second—170 species [146]. Longhorn beetles and jewel beetles are represented among invaders to a less extent than it might have been assumed according to a number of species in these families. Still, the representation of bark beetles, auger beetles, and horntails is disproportionately higher.

One of the obvious explanations for this disproportion is the confinement of the families to different zoogeographic regions. The non-native, secondary range of most studied invasive species is located in subtropical and temperate zones. Therefore, it seems logical to assume that the proportion of families widespread in the extratropical regions will be higher. However, these families’ greatest species diversity is concentrated in places with warmer climates [53,146,147,148], except for Siricidae [137]. Although it is hard to deny the role of zoogeographic patterns [10,149,150,151], we are forced to state that in this case, they do not allow us to understand the reason for the relatively high representation of some taxa in comparison with others. A possible exception is the Castniidae family, represented exclusively by tropical species [146] and predominantly Palaearctic and Nearctic Siricidae. However, both of these families are small and do not determine the taxonomic structure of invasive bark beetles and borers. 

The relatively high representation of Bostrychidae and Scolytinae among alien xylophagous insect species in general and invasive species, in particular, can be explained by their biological characteristics. Some of their biological characteristics help mitigate the negative impact of the Allee effect in the early phases of invasion (see Reproductive traits of invasive bark beetles and borers); others provide food for newly formed populations (see Insect-plant interactions).

### 4.2. Reproductive Traits of Invasive Bark Beetles and Borers

One of the fundamental biological patterns affecting alien species populations is the Allee effect [11]. According to this biological phenomenon, a population at low density faces difficulties in maintaining its size until the population increases to a specific density threshold [152]. Allee effect in an invasive species may affect the mate-finding mechanism, which has been well-studied for forest insects using the example of the gypsy moth *Lymantria dispar* (L.) [153,154]. 

To mitigate the Allee effect on the early stages of invasion, the mechanisms of opposite-sex conspecific individuals meeting and mating must exist [155]. Allee effect can also be weakened if species reproduce parthenogenetically or inbreed [10]. Parthenogenesis is typical of the tribe Xyleborini (Coleoptera: Curculionidae: Scolytinae), characterized by haplodiploid sex determination [156]. Xyleborini species, as well as *D. micans*, are also capable of inbreeding [53,151]. Indeed, the proportion of the tribe Xyleborini among alien species is much higher than would be expected [157], considering the number of species included in it (about 1200 [53]). Females often fly away once their sib-mate has fertilized them within galleries, which removes the problem of finding opposite-sex individuals [151,156,157] and reduces the number of individuals required for a successful introduction [98].

Another trait helping to avoid the impact of the Allee effect during the early phase of invasion is aggregation pheromones and long-range sex pheromones [10]. Both of these semiochemicals groups attract conspecific individuals. Aggregation pheromones act irrespective of gender; sex pheromones mitigate mate-finding [158]. Modeling showed that individuals’ concentration at the introduction and establishment stages contributes to increased population size, thereby ensuring its success [156]. Among the studied species, aggregation pheromones were found in several bark beetle species and long-range sex pheromones—in species of all families of beetles. Even though insect pheromones have been known since 1959, their prevalence in the class Insecta and its taxa is still insufficiently examined even for relatively well-studied bark beetles and longhorn beetles [53,148]. However, the fact that such species make up about 40% (18 species) of the total number of invasive bark beetles and wood borers indicates a serious advantage of pheromone communication. Remarkably, their list does not include any inbreeding species using only short-range pheromone communication, which ensures interaction within the nest and prevents outbred crossing with closely related species [103].

Allee effect in an invader population also depends on the reproductive rate [159], which is directly related to fecundity. For example, according to a model describing mosquito population dynamics, the increase in abundance is positively related to fecundity and inversely related to the strength of Allee effect (see Equation (2) in [160]). High fecundity is also a competitive advantage over native species [161,162]. As a matter of interest, invasive bark beetles and borers (especially of species belonging to the inbred-type) (Figure 4) are characterized by relatively low fecundity compared to insects in general [163]. In can be explained based on the results obtained in the study of native and invasive ichthyofauna of Central Europe [164]. Almost all Central Europe invasive species are typical K-strategists. They provide care for their offspring, compensating for reduced fecundity compared to native species. This pattern is also correct for the present study since low-fecund *inbred*-type insects are characterized by advanced parental care [53]. Life cycle also influences a population growth rate: the larger number of generations developed and started reproduction per unit of time, the more rapid population growth is.

### 4.3. Insect-Plant Interactions 

Along with the Allee effect, an essential factor for the newly established invasive population is host availability. Modeling an invasion process in imported wood storage has proved that the population establishment’s success depends on the amount of food available [159]. Although the model was created for specific storage conditions, it is based on the model of *Ips typographus* L. outbreak in forest areas [165] and is suitable for describing the processes occurring in stands. According to the model [165], the food available to bark beetles and borers includes dying and dead trees, which are unable to resist insect attacks and viable trees defense mechanisms of which can be overcome when aggressive insects population increases above a certain level.

Those invaders with no mechanisms to counteract plants’ defenses can only expand their host range by feeding on a wide range of trees and shrubs species [10]. Indeed, oligo- and polyphages predominate among the invasive bark beetles and borers’ species (see Characteristics of the studied invasive species). A significant part of polyphages is represented by species of the tribe Xyleborini, which typically develop on plants belonging to different families [166] due to their obligatory xylomycetophagy [53]. Different fungi species have different qualitative nutritional requirements of the feed substrate [167], and a wide diversity of symbionts allows ambrosia beetles to develop on taxonomically distant plant species [168] successfully. The same applies to other xylomycetophage species. 

A broad spectrum of host plants also characterizes species of the Bostrichidae family. Most members of this group (except *A. monachus*) damage dry wood, including finished wood products [24,25,26], as well as non-wood objects [169,170,171]. The mentioned feature also facilitates the spread of the Bostrichidae family’s species, expanding the range of possible invasion pathways (live plants, freshly harvested wood, any other wood materials, or the surface of containers and vehicles) [9].

Oligophagy and polyphagy are widespread in other families as well. As a general rule, species of xylophagous insects with the broader range of host species have the lower ability to develop on viable trees [172]. However, one often has to deal with this pattern violation by analyzing the studied species list (Table 1). Indeed, many oligophages and polyphages in their native habitat are associated with dying or recently dead trees [38,44,64,148,173]. Such species’ high impact outside their native range may be related to an association with pathogenic microorganisms (Table 6). Others (*L. festiva*, *A. glabripennis*, *P. rudis*, *R. rusticus*) seem to occupy trees weakened by drought [30,78,174] or other abiotic factors [45]. Finally, *C. lapathi* causes serious harm primarily when feeding on the North American species of willow and poplar [52,175], which have no resistance to this insect.

Some invasive species ensure population growth by overcoming host-plant resistance. As a rule, *aggressive*-type species are vectors of pathogens (Figure 2, Table 6) and use aggregation pheromones at the same time (Figure 2). The pathogenic microorganism makes it easier to overcome plant defense against herbivory, ensuring mass establishment within a short time [53,184]. The aggregation creates an additional load on a tree, forcing it to consume more resources for plant resistance to herbivory [185]. This triggers the two competing induced signaling pathways in plants: Salicylic acid against insect attacks and jasmonic acid against biotrophic and hemibiotrophic fungi (for example, *O. novo-ulmi* [186]) [187]. Activation of the jasmonic acid signaling pathway reduces the salicylic acid signaling pathway’s effectiveness, which prevents the formation of host plant resistance. The association between aggressive bark beetle species and phytopathogenic symbionts is not a prerequisite for the successful attacks on living trees [188,189]. However, bark beetle–fungus symbiosis weakens trees and expands the invader host plant range [53,190], which is crucial at the introduction stage. Critics of the idea that phytopathogenic fungi are critical for a successful attack rightly point out that insect colonization is completed before the pathogens have time to weaken the tree significantly [189]. However, many species repeatedly attack a tree before it becomes established [191,192] or gradually colonize one [72]. In both cases, trees are repeatedly subjected to infection as pathogens enter into plant tissues, often leading to a critical decline of tree vitality. At the final stage of an invasion, symbiotic interactions between insects and pathogens cause severe damage in forest stands, for instance, in *Polygraphus proximus-Grossmannia aoshimae-Abies sibirica* [83] and *Xyleborus glabratus-Raffalea lauricola-Persea borbonia* [193,194] systems. 

Host range expansion to healthy trees is also possible when native tree species do not have mechanisms to resist invasive pests’ attacks effectively. A typical example is *A. planipennis*, which is native to the Russian Far East and North-Eastern China. It successfully attacks North American and European ash species and causes mass damage in the USA, Canada, and the European part of Russia [195]. Remarkably, damage caused by *A. planipennis* in its natural range is much less [106]. Comparative analysis showed that defensive Far Eastern *Fraxinus* species have a set of specific alleles that provide adequate protection against *A. planipennis* [196]. These data are also confirmed at the physiological level [197].

### 4.4. Differences between Invasive and Non-Invasive Bark Beetles and Wood Borers

The results of factor analysis were unexpected. There was no association between invasiveness and the traits that define *inbred* strategy (in particular, xylomycetophagy and inbreeding) (Figure 6). A large number of species possessing these features demonstrate its great importance for establishing new populations [10], but do not guarantee the species will have a negative impact.

Two traits are associated with invasiveness: Fecundity and polyvoltinism (Figure 6). The high number of offspring and the ability to complete two or more life cycles per year (at least in regions with favorable climate) give bark beetles and borers the opportunity to increase their populations rapidly. Rapid population growth contributes to the cooperative effect [184] and overcoming natural enemy populations [172], leading to an outbreak. Notably, polyvoltine species are more prevalent in *inbred*-, *intermediate*- and *aggressive*-type, but high fecundity is more common in *polyphagous*-type (Figure 4). 

The relationship between imaginal aphagia and invasiveness is indirect in nature. Relatively high fecundity is typical for both invasive species and aphages. We suppose that short-lived aphagous insects have the opportunity to allocate resources to reproduction, which acts as a detriment to maintenance (see the principal scheme of resource allocation in [198]). The maturation feeding (or its absence) does not affect the potential to cause damage. 

It is easy to explain the role of pheromone communication and plant pathogens on invasiveness (Figure 6; for more details, see Insect-plant interactions). Long-range sex pheromones can help overcome Allee effect and facilitate cooperative effect in populations. It had been demonstrated in the phyllophagous species *Dendrolimus sibiricus* Chetverikov that the early stage of an outbreak is characterized by concentration of insects in the outbreak spot [172]. Apparently, the same mechanism mediated by long-range sex pheromones works in wood-boring invasive insect populations, which explained some *polyphagous* species’ invasiveness (Figure 2). Aggregation pheromones act similarly but also help overcome plant immunity by mass attacks, frequently in combination with plant pathogens [53]. Such a combination is typical for *aggressive*-type species (Figure 2).

We have no satisfactory explanation for the role of diet breadth in the ability to heavily damage trees and shrubs beyond the native range (Figure 6). Both mono- and polyphagy give some advantages in terms of the ability to cause environmental and socio-economic damage. Still, the nature of these advantages is unclear. 

Some of these traits (voltinism, host specificity, fecundity, parthenogenesis) are used in pest risk analysis procedures, but not entirely [199,200]. It should be considered that such traits as pheromones and symbiotic plant pathogens are necessary for a more precise prognosis of potential damage in the secondary area. 

In the non-invasive pool of species, there are specimens with no traits typical for invasive ones (for example, *A. angustulus, C. violaceum, C. piceae*). Interestingly, some potentially harmful species are included in this pool too. The most typical example of such species is the six-toothed bark beetle *I. sexdentatus*. This species is widely distributed throughout Extratropical Eurasia [64,201], but alien in Great Britain [16]. In our opinion, two factors prevent damage from *I*. *sexdentatus*. First, Scots pine, the main European host plant of *I. sexdentatus*, occupies a relatively small area in Great Britain [202], especially in its southern part, where established populations of this species are observed at present [203]. Second, the climate of Great Britain is humid with a relatively cool summer [204]. The six-toothed bark beetle is thermophilic species [64], and its outbreaks are related to droughts [205], which is typical for many other bark beetles [53]. Perhaps, the climatic conditions of the secondary area might prevent damage by *I. sexdentatus*. 

In practice, the biological traits are not the only predictors of a potential invasion. Other factors should also be examined, for example, climatic conditions of pest risk analysis area [199] (or, as an approximation, biogeographic origin [151]), host range [151], and pheromone communication of native species that can complicate the establishment of alien species with similar semiochemicals [206,207]. 

### 4.5. Future Prospects

The features contributing to bark beetles and borers’ invasion success have been studied to varying degrees. Their diet breadth can be considered thoroughly investigated, at least for extratropical faunas. The same applies to the rate of development. Inbreeding and parthenogenesis have been studied sufficiently in the Xyleborini to recognize their usefulness in population establishment. In our opinion, since they may not have as great of an impact on their potential for damage as some other traits, there is no need to conduct any additional research. Although the data on many species’ fecundity requires clarification, information is often available for closely related taxonomic groups, which allows plausible assumptions to be made. 

Despite significant advances, there are numerous knowledge gaps in pheromone communication in insects, including bark beetles and borers. Even for such a well-studied group as the subfamily Scolytinae, there is no data on aggregation pheromones presence/absence for many species [70], or the data are contradictory (an example of *D. micans*, see [53,54]). However, aggregation pheromones are widespread in the taxon and essential for insects and their host plants’ interactions [53]. Knowledge of sex pheromones is also fragmentary (an example of the Cerambycidae family, see [148]) or insufficient [28]. Reasons for the persistence of such knowledge gaps include the need for expensive equipment and laborious research. Nevertheless, information on insect pheromones is crucial for an adequate pest risk assessment. 

There also is a research gap in pathogens associated with wood-boring insects. As in the case with pheromone identification, such work requires significant resources. Therefore, laboratory experiments on the possibility of insect-pathogen association may not be confirmed by field experiments [87]. The regional specificity of such associations creates additional difficulties. For example, the species of *H. ligniperda*-associated fungi, belonging to the genera *Ophiostoma* and *Leptographium*, differs significantly in New Zealand [67], Poland [208], and Ukraine [66]. It is likely leading to a lack of essential information. Such differences may also be the consequence of differences in research methods or the fungal communities’ specificity formed on different tree species. Finally, the pathogenicity of some associates remains controversial (see an example of *O. minus* in [60]), which additionally complicates pest risk assessment. There is also a problem predicting the consequences of forming new associations between insects and pathogens in the invaded site [83,177]. 

Alien species pose a severe threat if evolutionarily developed native plant defenses are not effective against them [196,209] or their associated pathogens [181,191,210]. Experiments on the interaction of potential invaders and their host species outside the primary insect range carry unreasonably high risks. However, such studies are possible using sentinel trees planted outside their native range (for example, in urban planting or botanical gardens) [211]. Such research is particularly valuable for species that do not cause significant damage in their native habitat [212]. We are not aware of any targeted attempts at pest risk assessment using sentinel trees for bark beetles and borers, excluding the study in [213]. However, similar studies were carried out in the secondary range of such insects as *P. proximus* [214] and *A. planipennis* [195,214] where both insects and plants were exotic. 

Another issue is the ability of bark beetles and borers to overwhelm tree defense mechanisms. The aggressiveness of these species is directly linked to their tolerance to host defenses [53]. There are some advances in the understanding of this problem [215,216], but our knowledge in this field is still insufficient. 

## 5. Conclusions

Forty-seven species were considered invasive according to the data on worldwide fauna of alien bark beetles and borers, as well as on the local faunas of the European part of Russia, the USA, China, and New Zealand. The taxonomic analysis of regional faunas at the family level showed their significant similarity, except for China. Cerambycidae and Curculionidae predominate in all regions for which there are complete data on the fauna of alien bark beetles and borers. 

Feeding on the inner-bark is the most common among the invasive species (57.4%). Many species feed on wood (31.9%); some are xylomycetophagous (29.8%). The food spectra of different species vary from monophagy to polyphagy. Maturation feeding is known for 74.5% of species. Inbreeding is known for 25.5% of invasive bark beetles and borers, most of which are capable of parthenogenesis. Fecundity varies from 25 to 110 eggs; the median value for all species is 83 eggs. Association with plant-pathogenic microorganisms is known for 34.0% of species. Aggregation pheromones were found in 25.5% of species, long-range sex pheromones—in 12.8%. The typical development rate is several generations (63.8%) or one generation per year (66.0%). Less often, development takes several years (34.0%). 

We identified four invasion strategies using multidimensional analysis. The *polyphagous* strategy is characterized by consuming a broad spectrum of host species, feeding on wood during the larval stage, and relatively high fecundity. Many species, using this strategy, produce long-range sex pheromones. Long-term development (one generation per year or developing over many years) of this strategy’ species prevent their establishment in the secondary range. A typical *intermediate* strategy representative is a monophagous or oligophagous, feeding on inner-bark at the larval stage, having medium fecundity, and one or several generations per year. Aggregation pheromones and associated plant pathogens provide the success of species with the *aggressive* strategy. This strategy’s other traits are monophagy, feeding on inner-bark, and a short life cycle (one or several generations per year). The *inbred* strategy is characterized by a set of features specific to the tribe Xyleborini, namely inbreeding, parthenogenesis, polyvoltinism, xylomycetophagy, flightless males, and polyphagy. Many of the *inbred* species are associated with pathogenic fungi. At the same time, they are distinguished by low fecundity and the absence of long-range sex pheromones. However, many species combine features of two or more invasion strategies, which leads to numerous transitions between them (excluding the *inbred* strategy). It should be mentioned that these results are preliminary because of the incomplete biological traits data for some species. 

It is important to note that not all of these traits are necessary to cause damage to native trees and shrubs species. The most important ones are high fecundity (typical for *polyphagous* and, to a lesser extent, *intermediate* strategies), polyvoltinism (all but *polyphagous*), symbiotic plant pathogens presence (*aggressive* and, to a lesser extent, *inbred*), long-range sex (*polyphagous*) and aggregation pheromones (*aggressive*). 

It is necessary to understand which bark beetles and borers’ biological traits are predictors of invasiveness to improve pest risk assessment. Therefore, it is crucial to focus on the potentially dangerous species with phytopathogenic symbionts, long-ranged or aggregation sex pheromones. However, further research on insect chemical communication and borer association with pathogenic microorganisms is necessary for more accurate forecasting. Studies using sentinel trees might help predict how alien species will interact with dendroflora in the secondary area. 

## Figures and Tables

**Figure 1 insects-12-00367-f001:**
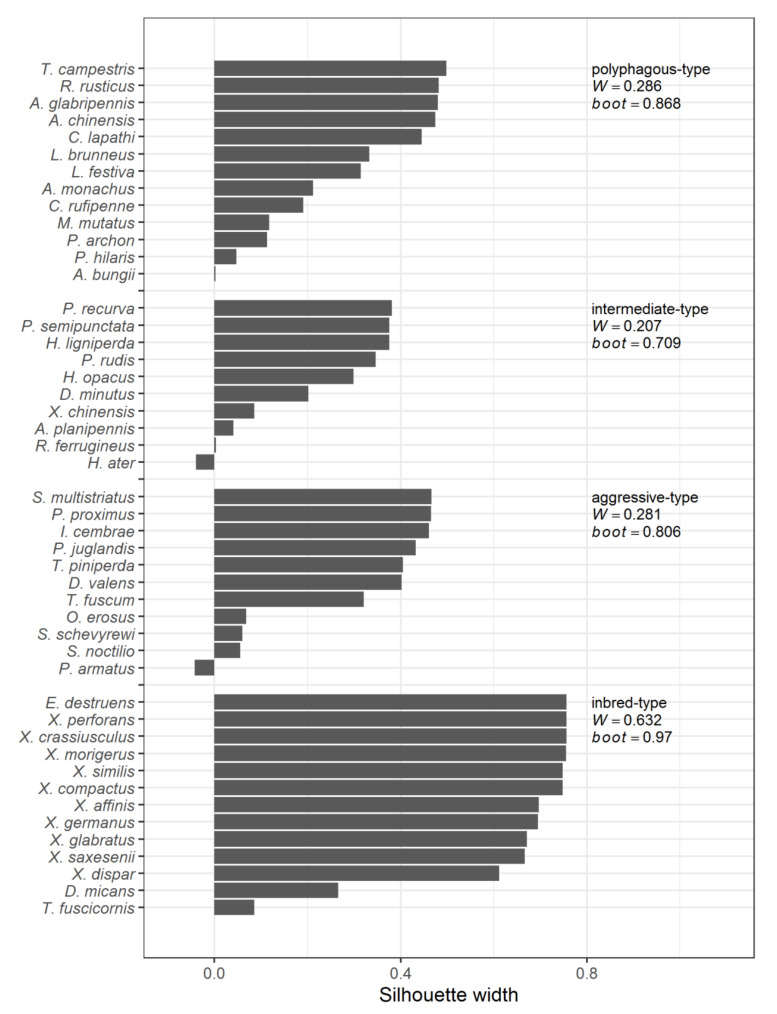
Groups of species identified by PAM clustering and characteristics of their compactness (*W*) and stability (*boot*).

**Figure 2 insects-12-00367-f002:**
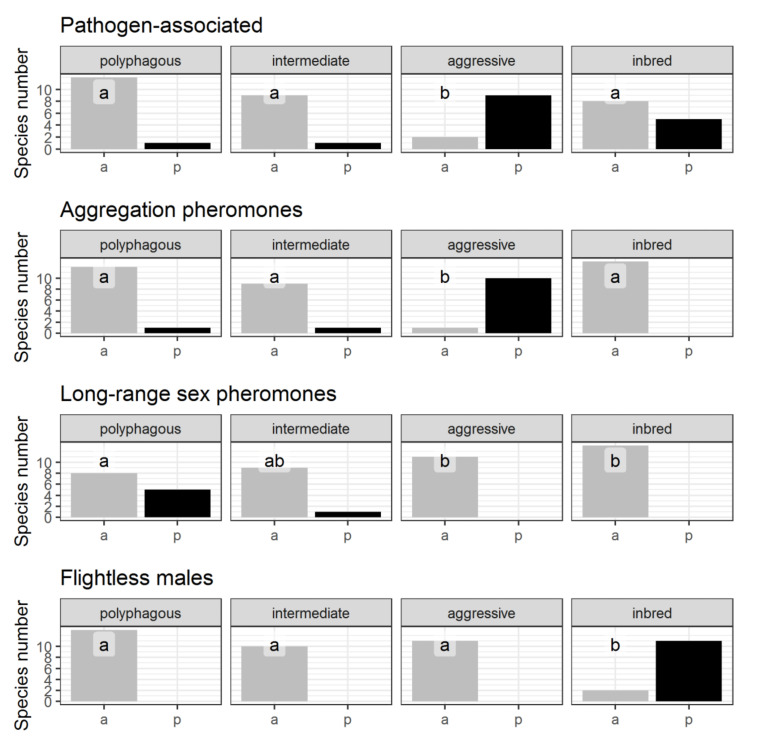
Biological traits characterizing aggressive behavior, communication, and the ability to fly. Axis abscissa label indicates the presence (p) or absence (a) of the feature. The same letters on the graphs indicate there are no differences between strategies at *p* ≤ 0.05.

**Figure 3 insects-12-00367-f003:**
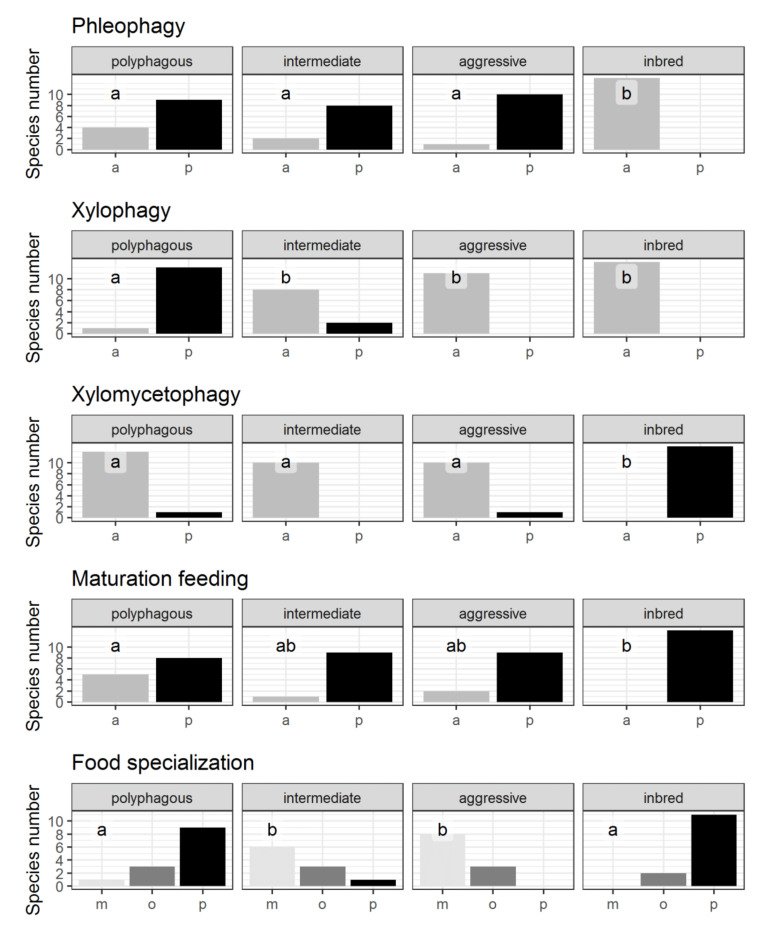
Biological traits characterizing nutrition. Axis abscissa label for binary characteristics indicates the presence (p) or absence (a) of the feature. Axis abscissa label for food specialization—monophagy (m), oligophagy (o), and polyphagy (p). The same letters on the graphs indicate no differences between strategies at *p* ≤ 0.05.

**Figure 4 insects-12-00367-f004:**
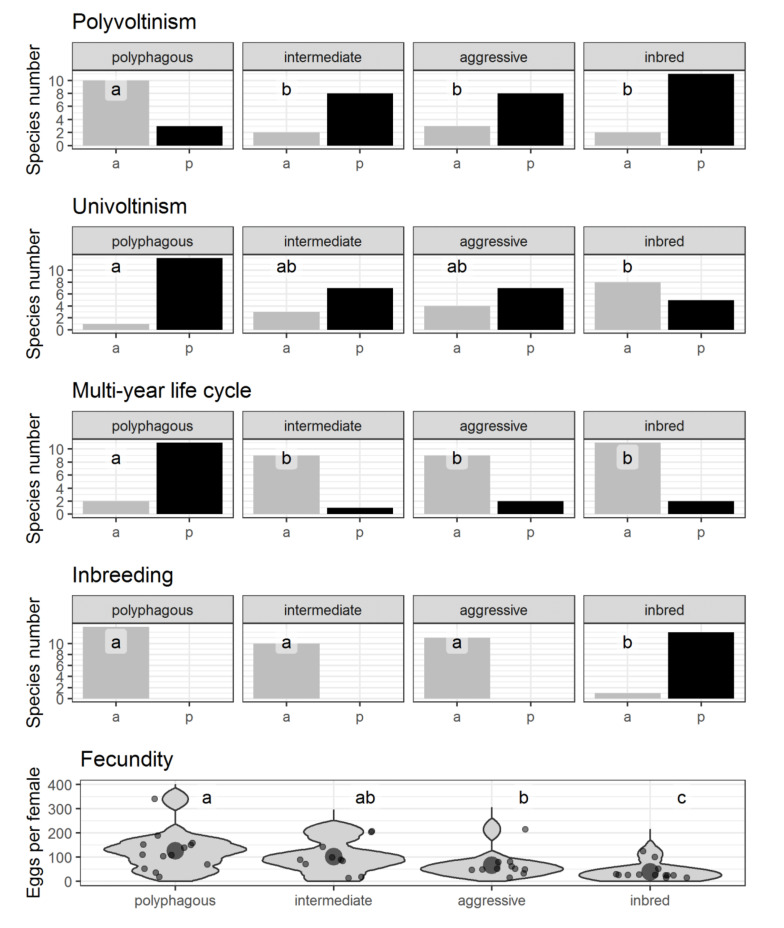
Biological traits characterizing reproduction. Axis abscissa label for binary characteristics indicates the presence (p) or absence (a) of the feature. The grey area’s width for fecundity indicates the probability density; dots show the species’ fecundity; circles represent the group’s mean fecundity. The same letters on the graphs indicate no differences between strategies at *p* ≤ 0.05.

**Figure 5 insects-12-00367-f005:**
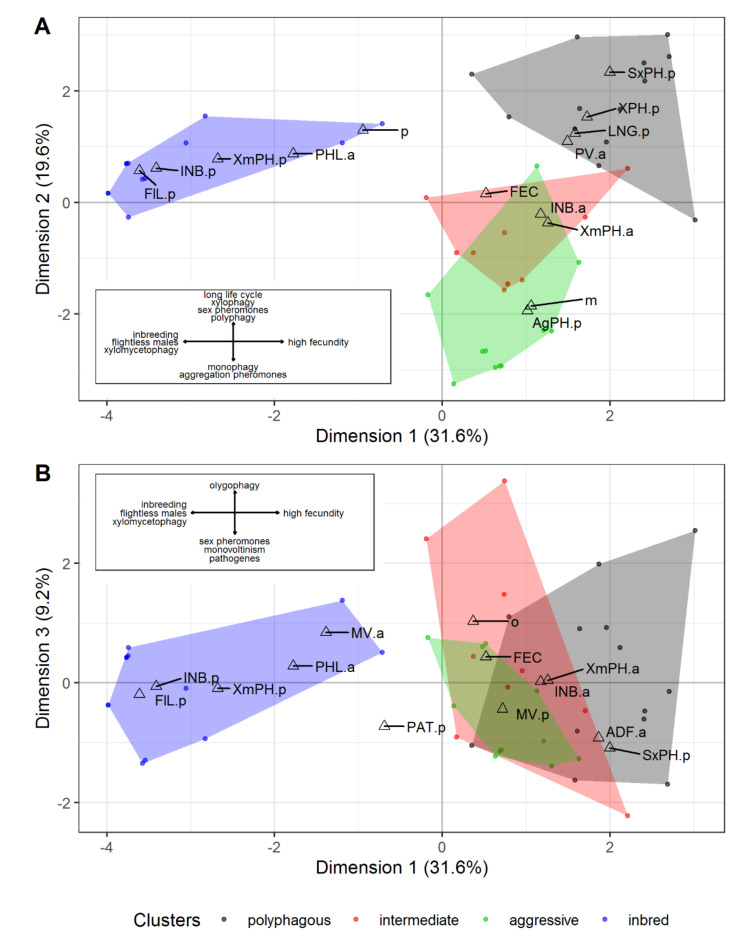
Factor analysis of invasive species traits results: (**A**)—Dimensions 1, 2, (**B**)—Dimensions 1, 3. Triangles indicate the position of biological traits with the highest factorial loads in the factor space: PHL, XPH, XmPH—phleo-, xylo- and xylomycetophagy; INB—inbreeding; FEC—fecundity; PAT—plant pathogens; SxPH, AgPH—long-distant sex and aggregation pheromones; FlL—flightless males; PV, MV and LNG—mono-, polyvoltinism and long (multi-year) life cycle; m, o, p—mono-, oligo- and polyphagy. For binary characteristics, .p and .a indicate the presence or absence of the feature, respectively. Dots indicate the position of species in the factor space. The insets show the distribution of species with corresponding features within the coordinate plane. A variance explains the percentage in axis labels.

**Figure 6 insects-12-00367-f006:**
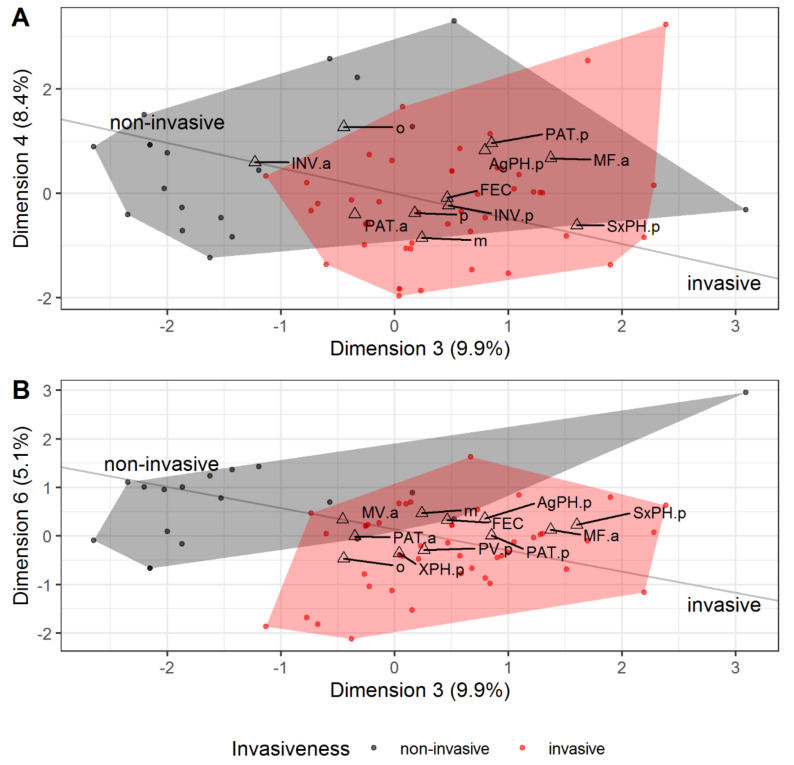
Factor analysis of invasive vs. non-invasive species traits results: (**A**)—Dimensions 3, 4, (**B**)—Dimensions 3, 6. The additional axis on the coordinate plane shows the direction from non-invasive to invasive species. MF—maturation feeding; for more details, see Figure 5.

**Table 1 insects-12-00367-t001:** Taxonomic identification of studied insect species and sources of information.

Species	Family	Main Sources	Additional Sources
*Apate monachus* Fabricius	Bostrychidae	[6,16]	[21,22]
*Dinoderus minutus* (Fabricius)	Bostrychidae	[9,14,16]	[23]
*Lyctus brunneus* (Stephens)	Bostrychidae	[9,16]	[24,25,26,27]
*Agrilus planipennis* Fairmaire	Buprestidae	[5,6,9,14,16]	[28]
*Lamprodila festiva* (Linnaeus)	Buprestidae	[9]	[29,30]
*Paysandisia archon* (Burmeister)	Castniidae	[6,14,16]	[31,32]
*Anoplophora chinensis* (Forster)	Cerambycidae	[6,16]	[33,34]
*Anoplophora glabripennis* (Motschulsky)	Cerambycidae	[5,6,14,16]	[35]
*Aromia bungii* Faldermann	Cerambycidae	[6,14]	[36,37]
*Callidiellum rufipenne* (Motschulsky)	Cerambycidae	[5,6,9,14,16]	[38]
*Phoracantha recurva* Newman	Cerambycidae	[5,6,14,16]	[39,40]
*Phoracantha semipunctata* (Fabricius)	Cerambycidae	[5,6,14,16]	[39,40]
*Psacothea hilaris* (Pascoe)	Cerambycidae	[6,16]	[41,42,43]
*Rusticoclytus rusticus* (Linnaeus)	Cerambycidae	[6]	[44,45]
*Tetropium fuscum* (Fabricius)	Cerambycidae	[14]	[46]
*Trichoferus campestris* (Faldermann)	Cerambycidae	[6,9]	[47,48]
*Xylotrechus chinensis* Chevrolat	Cerambycidae	[6]	[44,49,50]
*Anisandrus dispar* (Fabricius)	Curculionidae	[14]	
*Cryptorrhynchus lapathi* Linnaeus	Curculionidae	[5,6]	[51,52]
*Dendroctonus micans* (Kugelann)	Curculionidae	[14,16]	[53,54]
*Dendroctonus valens* LeConte	Curculionidae	[14]	[53,55,56]
*Euwallacea destruens* (Blandford)	Curculionidae	[14]	[53,57]
*Hylastes ater* (Paykull)	Curculionidae	[14,15,16]	[58,59,60,61]
*Hylastes opacus* Erichson	Curculionidae	[5]	[62,63,64]
*Hylurgus ligniperda* (Fabricius)	Curculionidae	[5,15]	[65,66,67]
*Ips cembrae* (Heer)	Curculionidae	[14,16]	[68,69]
*Megaplatypus mutatus* (Chapius)	Curculionidae	[14,16]	[70,71,72,73,74]
*Orthotomicus erosus* (Wollaston)	Curculionidae	[5,14]	[60,70,75]
*Phloeosinus armatus* (Reitter)	Curculionidae	[5,16]	[20,76]
*Phloeosinus rudis* Blandford	Curculionidae	[6,16]	[77,78]
*Pityophthorus juglandis* Blackman	Curculionidae	[6]	[79,80,81]
*Polygraphus proximus* Blandford	Curculionidae	[6,9,16]	[82,83,84]
*Rhynchophorus ferrugineus* (Olivier)	Curculionidae	[6]	
*Scolytus multistriatus* (Marsham)	Curculionidae	[5,15]	[7,64,85,86]
*Scolytus schevyrewi* Semenov	Curculionidae	[5]	[87,88]
*Tomicus piniperda* (Linnaeus)	Curculionidae	[5]	[70,89,90,91]
*Xyleborinus saxesenii* (Ratzeburg)	Curculionidae	[14,15,16]	
*Xyleborus affinis* Eichhoff	Curculionidae	[16]	[92,93]
*Xyleborus glabratus* Eichoff	Curculionidae	[5]	[94]
*Xyleborus perforans* (Wollaston)	Curculionidae	[14,16]	
*Xyleborus similis* Ferrari	Curculionidae	[14]	[95]
*Xylosandrus compactus* (Eichhoff)	Curculionidae	[6,14]	[96,97]
*Xylosandrus crassiusculus* (Motschulsky)	Curculionidae	[6,14,16]	[98,99]
*Xylosandrus germanus* (Blandford)	Curculionidae	[6,9,14,16]	[100]
*Xylosandrus morigerus* (Blandford)	Curculionidae	[14,16]	
*Sirex noctilio* Fabricius	Siricidae	[5,14,15,16]	[101,102,103]
*Tremex fuscicornis* (Fabricius)	Siricidae	[14]	[101,104]

**Table 2 insects-12-00367-t002:** Studied aspects of the biology of invaders.

Trait	Units/Categories
Type of nutrition	Phloeophagy, Xylophagy, Xylomycetophagy
Maturate feeding	Yes, No
Food specialization	Monophagy, Oligophagy, Polyphagy
Fecundity	Number of eggs produced by a female
Parthenogenesis	Yes, No
Inbreeding	Yes, No
Life cycle	Polyvoltine, univoltine, Multi-year life cycle
Aggregation pheromones	Yes, No
Long-range pheromones	Yes, No
Associated pathogen	Yes, No
Ability to fly	Yes, No

**Table 3 insects-12-00367-t003:** Taxonomic identification of studied non-invasive insect species and sources of data.

Species	Family	Main Sources	Additional Sources
*Agrilus angustulus* (Illiger)	Buprestidae	[16]	[111,112]
*Agrilus sulcicollis* Lacordaire	Buprestidae	[5]	[113]
*Melanophila acuminata* (De Geer)	Buprestidae	[16]	[114,115,116]
*Callidium violaceum* (Linnaeus)	Cerambycidae	[5]	[117,118]
*Hylotrupes bajulus* (Linnaeus)	Cerambycidae	[5]	[118,119,120]
*Morimus asper* (Sulzer)	Cerambycidae	[16]	[121,122]
*Saperda candida* Fabricius	Cerambycidae	[6,16]	[123,124]
*Saperda populnea* Linnaeus	Cerambycidae	[5,14]	[125]
*Tetropium castaneum* (Linnaeus)	Cerambycidae	[5,14]	[46,126,127]
*Tetrops praeusta* (L.)	Cerambycidae	[5]	[127]
*Zeuzera pyrina* (Linnaeus)	Cossidae	[5,14]	[128]
*Anisandrus maiche* (Kurentsov)	Curculionidae	[9]	[129]
*Cryphalus piceae* (Ratzeburg)	Curculionidae	[16]	[64,130,131]
*Hylurgops palliatus* (Gyllenhal)	Curculionidae	[5,14]	
*Ips sexdentatus* Boerner	Curculionidae	[5]	[64,132,133,134]
*Pityogenes quadridens* (Hartig)	Curculionidae	[5]	[64,135]
*Scolytus rugulosus* (Müeller)	Curculionidae	[5]	[64,136]
*Uroceurs gigas* (Linnaeus)	Siricidae	[5]	[101,137]

**Table 4 insects-12-00367-t004:** The data has been obtained for related species or statistically modelled (s.m.).

Species	Maturation Feeding	Life Cycle	Fecundity	Ability to Fly
*X*. *chinensis*	*Xylotrechus pantherinus* (Savenius), *X*. *arvicola* (Oliver)			
*H*. *opacus*	*Hylastes ater* (Paykull)	*H*. *ater*	s.m.	*H*. *ater*
*M. acuminata*	*Melanophila unicolor* Gory			
*C. piceae*	*Cryphalus scopiger* Berger		s.m.	
*L. brunneus*		*Lyctus* spp.		
*A*. *chinensis*			*Anoplophora malasiaca* Thomson	
*P*. *rudis*			*Phloeosinus bicolor* (Brullé)	
*C. violaceum*	s.m.		*Callidielum aeneum* (De Geer), *C*. *coriaceum* Paykull	
*T. praeusta*			*Tetrops elaeagni* Plavilsthikov	
*P. quadridens*			*Pityogenes chalcographus* (Linnaeus)	
*L*. *festiva*			s.m.	
*T*. *fuscicornis*	s.m.		s.m.	
*E.destruens*			s.m.	
*X.perforans*			s.m.	
*X*. *glabratus*	s.m.		s.m.	
*A*. *angustulus*			s.m.	
*A*. *sulcicollis*			s.m.	
*A*. *maiche*			s.m.	
*U*. *gigas*			s.m.	
*M*. *mutatus*	s.m.			

**Table 5 insects-12-00367-t005:** Distribution of alien bark beetles and borers’ species by families within a region and for the entire set of studied species, as well as the distribution of alien species, generalized for regional faunas.

Family	The European Part of Russia ^1^	China	USA	New Zealand	All Invasive Bark Beetles and Borers ^2^	All Alien Bark Beetles and Borers ^3^
Acanthocnemidae		1				1
Agonoxenidae			1			1
Anobiidae		1	1			2
Bostrychidae	4	2			3	5
Buprestidae	2	3	6		2	9
Cerambycidae	2	19	13	9	11	36
Cossidae			1			1
Curculionidae	6	22	43	11	28	78
Kalotermitidae		1				1
Rhinotermitidae		1	1			2
Sesiidae			2			2
Siricidae		3	2	1	2	5
Castniidae		1			1	1

^1^ Only insects of the order Coleoptera. ^2^ From the species included in the present research. ^3^ Combined data [5,6,9,15].

**Table 6 insects-12-00367-t006:** Bark beetles and borers as pathogen vectors and their associated pathogens.

Insect	Pathogen	Data Source
*T*. *fuscum*	*Grosmannia piceiperda* (Rumbold) Goid.	[126]
*D*. *valens*	*Leptographium terebrantis* S.J. Barras & T.J. Perry, *Leptographium procerum* (W.B. Kendr.) M.J. Wingf.	[14,176,177]
*H*. *ater*	*Ophiostoma minus* (Hedgc.) Syd. & P. Syd.,	[60,178]
*I*. *cembrae*	*Ceratocystis laricicola* Redfern & Minter	[69]
*M*. *mutatus*	*Raffaelea santoroi* Guerrero, *Fusarium solani* (Mart.) Sacc.	[74,179,180,181]
*P*. *armatus*	*Seiridium cardinale* (W.W. Wagener)	[20]
*P*. *juglandis*	*Geosmithia morbida* M. Kolařík, Freeland, C. Utley & Tisserat	[81]
*P*. *proximus*	*Grosmannia aoshimae* (Ohtaka et Masuya) Masuya et Yamaoka	[83]
*S*. *multistriatus*	*Ophiostoma ulmi* (Buisman) Nannf.	[7]
*T*. *piniperda*	*Leptographium wingfieldii* M. Morelet, *O*. *minus*	[91]
*X. affinis*	*Raffaelea lauricola* T.C. Harr., Fraedrich & Aghayeva	[93]
*X*. *glabratus*	*R*. *lauricola*	[168]
*X*. *similis*	*F*. *solani*	[95]
*X*. *compactus*	*F*. *solani*	[96]
*X*. *germanus*	*F*. *solani*	[182]
*S*. *noctilio*	*Amylostereum areolatum* (Chaillet ex Fr.) Biodin	[183]

## Data Availability

The data presented in this study are available on request from the corresponding author.

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
