# Peer review of "Biological Strategies of Invasive Bark Beetles and Borers Species"

_insects, 2021, doi:10.3390/insects12040367_

Round 1

Reviewer 1 Report

The edited manuscript is much improved. I have added some edits to clarify language and syntax in the pdf. I believe the paper will be appropriate for publication once these changes are completed. 

Author Response

Dear collegue, we're grateful to You for Your time, comments and advices. We've tried to correct language and syntax disadvantages in the text. 

With best wishes,

Denis Demidko & co-authors

Reviewer 2 Report

The paper has been definitely improved by the authors. I thank them for having addressed most of my previous comments and I think that the paper can now be considered for publication. I only recommend the authors to have the paper read by a native English speaker as there are several sentences that has to be improved. Pay also attention to scientific name of species listed in Tables.

Author Response

Dear collegue, we're grateful to You for Your time, comments and advices.

We've tried to correct language and syntax disadvantages in the text. Scientific names from tables have been checked and one mistake (Ploeosinus) have been fixed. 

With best wishes,

Denis Demidko & co-authors

This manuscript is a resubmission of an earlier submission. The following is a list of the peer review reports and author responses from that submission.

Round 1

Reviewer 1 Report

This manuscript attempts to classify characteristics of invasive wood boring beetles into categorical strategies (inbred, polyphagous, fast-developing, and aggressive) for use in predicting what species are most likely to become invasive. The data set selection and explanation of character states was well defined.

The method for using information from closely related species to estimate missing data is appropriate, but caution should be taken with any conclusions drawn from incomplete data. 

The authors need to be consistent in their use of terms. There are places where they change the name of the inbred/inbreeding category. Stick to one name. Also, italicize the category name throughout the text or at first mention. There appears to be random use of italics in the text.  

I would like to see a stronger conclusion paragraph that more specifically explains how this information can be used for regulatory purposes.  It would also be useful for the authors to address whether there are alien insects with these same combinations of characteristics that are NOT invasive. If so, why are they not invasive and could they become invasive in different areas of the world? 

The wrong symbol is used for apostrophe throughout the text. That needs to be corrected. 

There are missing references in the text. Be sure all references are present and cited appropriately. 

The manuscript requires thorough English grammar editing prior to publication. 

Additional comments are on the pdf. 

Author Response

Dear collegue, Thank You for Your time and advices! Here I try to answer to Your comments.

...but caution should be taken with any conclusions drawn from incomplete data. 

This disclaimer have been added into L640-641.

The authors need to be consistent in their use of terms. There are places where they change the name of the inbred/inbreeding category. Stick to one name. Also, italicize the category name throughout the text or at first mention. There appears to be random use of italics in the text.  

This drawbacks have been corrected.

I would like to see a stronger conclusion paragraph that more specifically explains how this information can be used for regulatory purposes. 

We have added our opinion about this matter to L537-540 (Discussion) and L648-650 (Conclusion).

It would also be useful for the authors to address whether there are alien insects with these same combinations of characteristics that are NOT invasive. If so, why are they not invasive and could they become invasive in different areas of the world? 

Sadly, but we could not consider this issue as close as we want. However, we analyze the example of such species (L543-552). 

The wrong symbol is used for apostrophe throughout the text. That needs to be corrected. 

There are missing references in the text. Be sure all references are present and cited appropriately.

Thank You; these disadvantages are corrected.

The manuscript requires thorough English grammar editing prior to publication. 

We intend use MDPI English Editing service before publishing. 

It would help the reader to describe how you are defining the difference between alien and invasive. 

Thank You very much. This clarification have been added to L49-51. 

Other minor comments from .pdf are corrected.

Best wishes,

Denis Demidko

Reviewer 2 Report

The paper by Demidko et al. is potentially interesting for many readers because i) it is focused on a very important group of exotic species, i.e., wood-boring beetles, and ii) it aims to understand which are the characteristics that make certain species more invasive than others. Despite its potential interest, I think that it has to be largely improved before possible publication. I have a number of major issues that I recommend the authors to consider before resubmitting the paper and that I hope can help the authors to improve it. I instead reported only a few minor comments because I think that they are not needed at this stage.

Major omments:

i) It is really difficult for the reader to understand which is the main aim of this study. The authors report that “There are features considered to determine some species` invasion success compared to others” and this makes me think that the authors want to understand why certain species are more successful invaders than others. However, the authors consider only a subset of the pool of alien species present in the different countries. In order to really understand why certain species have a higher success than others, it would be important to add to the analysis also exotic species that are not invasive, and compare characteristics of invasive vs. not-invasive species. This would allow to answer the question why certain species are invasive and others not.

ii) More details are needed regarding the criteria used to select the species retained for the analysis. "Invasive species" is a really broad term and the authors have to clearly explain why they included certain species and not others. For example, why Ambrosiodmus minor is not included in the list for USA (see Hulcr et al. 2021)? And Euwallacea fornicatus (many papers show its impact in the USA and other countries)? The authors should add a paragraph where they thoroughly explain the criteria they used for selecting species to analyze and how they searched for information about the different species that allowed them to classify the latter as invasive or not. In addition, they should carefully revise the list of species and add all the missing ones. In my opinion relying solely on the CABI website is not the right procedure because they are missing a number of important species. Thus, I recommend to better look into the literature because there are many publications reporting the list of alien species in the different countries that can complement the CABI website. 

iii) The authors included only certain countries and not others without clearly explain the reason for this choice. They reported that “we used data on their fauna in the regions where it is represented by a significant number of species and is particularly well studied”. Then why to exclude countries in Central Europe? Wood-boring beetle fauna is very well studied in most European countries thus I don’t see why not to include them. Native and exotic wood-boring beetle fauna is really well studied in France (see Barnouin et al. 2020, Bark beetles and pinhole borers recently or newly introduced to France (Coleoptera: Curculionidae, Scolytinae and Platypodinae), Italy (see the book chapter by Rassati et al. 2016), UK (see Inward et al. 2020 Three new species of ambrosia beetles established in Great Britain illustrate unresolved risks from imported wood) and others. This is an important point that should be addressed by including the latter countries and the related species to this study. This would allow you to include some important species to the list, such as Pithophtorus juglandis. 

iv) I recommend to add a table to the Supplementary Materials reporting the biological traits that you assigned to each species. This way a reader can check if the info are correct or not. 

v) The discussion is too long and should be shortened. In addition you are not considering a key aspect of the whole story, that is the effect of abiotic variables on wood-boring beetle invasions (see the paper by Marini et al. 2011 Exploring associations between international trade and environmental factors with establishment patterns of exotic Scolytinae: Rassati et al. 2016 Bark and ambrosia beetles show different invasion patterns in the USA; Lantschner et al. 2020 Drivers of global Scolytinae invasion patterns). This part should be included in the text, either in the introduction or in the discussion.

Minor comments:

Line 73-74: what do you mean with higher rank?

Line 82-83: it is not clear to me where these data are from. Opening the link present in the reference 14, it is possible to notice that it refers to the general home page of CABI. I think that more details are needed here.

Line 93-96: this is unclear. So how did you classify Xyleborine species?

Line 380-398: there is another important point that should be discussed here, which is the pheromone-free space hypothesis. That is, a given species is able to establish in a novel country only when it does not share pheromones with native species (this is very common in longhorn beetles) or it has a seasonal or diel phonology different to that of the native species with which it shares the sex-pheromone (see Hanks, L.M. and Millar, J.G. (2018) Conservation of pheromone chemistry within the Cerambycidae on a global scale, and implications for invasion biology and Rassati et al. 2020. Response of native and exotic longhorn beetles to common pheromone components provides partial support for the pheromone-free space hypothesis).

Table 1: Xyleborus dispar is now Anisandrus dispar.

Table 3: this table and the information reported in it comes out of the blue. In addition they are not complete and should be explained.

Author Response

Dear collegue, Thank You for Your time and comments. Here we try to answer to Your criticism. 

In order to really understand why certain species have a higher success than others, it would be important to add to the analysis also exotic species that are not invasive, and compare characteristics of invasive vs. not-invasive species. 

It have been implemented (see Section 3.5).

"Invasive species" is a really broad term and the authors have to clearly explain why they included certain species and not others. For example, why Ambrosiodmus minor is not included in the list for USA (see Hulcr et al. 2021)? And Euwallacea fornicatus (many papers show its impact in the USA and other countries)? The authors should add a paragraph where they thoroughly explain the criteria they used for selecting species to analyze and how they searched for information about the different species that allowed them to classify the latter as invasive or not.

It is really complicated problem, and it should be given two answers to it. First, the main criteria of including described in L95-97. We consider the given species as invasive if local specialists believe that it is dangerous in the secondary area. The second criteria of including (or excluding) species into consideration is completeness of data about its biological traits. Such data is very incomplete even for such well-known and widely distributed taxa as IpsAgrilusCerambyx, Siricidae as a whole etc. Even for a lot of included species we used some tricks for fill the gaps in our knowledge (see L152-156). Then, we forced to exclude such harmful species as Efornicatus because of lack of information (L156-157). 

In addition, they should carefully revise the list of species and add all the missing ones. In my opinion relying solely on the CABI website is not the right procedure because they are missing a number of important species. Thus, I recommend to better look into the literature because there are many publications reporting the list of alien species in the different countries that can complement the CABI website. 

We used not only CABI data but also some regional lists (L94-99, Table 1). There is quite a lot of species which are not mentioned in CABI data sheets but included in our research. 

Then why to exclude countries in Central Europe? Wood-boring beetle fauna is very well studied in most European countries thus I don’t see why not to include them. 

Thank You for Your advise. Instead of processing of country-wide lists we have use DAISY database which includes most of European and Mediterranean species. It should be mentioned that the lists of DAISY, European part of Russia and USA are largely overlapped (see Table 1), but we add some new species in our set of invaders, and it is very useful. 

I recommend to add a table to the Supplementary Materials reporting the biological traits that you assigned to each species. 

We have prepared this Supplementary

v) The discussion is too long and should be shortened. In addition you are not considering a key aspect of the whole story, that is the effect of abiotic variables on wood-boring beetle invasions (see the paper by Marini et al. 2011 Exploring associations between international trade and environmental factors with establishment patterns of exotic Scolytinae: Rassati et al. 2016 Bark and ambrosia beetles show different invasion patterns in the USA; Lantschner et al. 2020 Drivers of global Scolytinae invasion patterns). This part should be included in the text, either in the introduction or in the discussion.

Minor comments:

Line 73-74: what do you mean with higher rank?

Line 82-83: it is not clear to me where these data are from. Opening the link present in the reference 14, it is possible to notice that it refers to the general home page of CABI. I think that more details are needed here.

Line 93-96: this is unclear. So how did you classify Xyleborine species?

Line 380-398: there is another important point that should be discussed here, which is the pheromone-free space hypothesis. That is, a given species is able to establish in a novel country only when it does not share pheromones with native species (this is very common in longhorn beetles) or it has a seasonal or diel phonology different to that of the native species with which it shares the sex-pheromone (see Hanks, L.M. and Millar, J.G. (2018) Conservation of pheromone chemistry within the Cerambycidae on a global scale, and implications for invasion biology and Rassati et al. 2020. Response of native and exotic longhorn beetles to common pheromone components provides partial support for the pheromone-free space hypothesis).

Table 1: Xyleborus dispar is now Anisandrus dispar.

Table 3: this table and the information reported in it comes out of the blue. In addition they are not complete and should be explained.
